# Trogocytosis-associated cell to cell spread of intracellular bacterial pathogens

**Shaun Steele\*, Lauren Radlinski, Sharon Taft-Benz, Jason Brunton, Thomas H Kawula\***

University of North Carolina at Chapel Hill, Chapel Hill, United States

**Abstract** Macrophages are myeloid-derived phagocytic cells and one of the first immune cell types to respond to microbial infections. However, a number of bacterial pathogens are resistant to the antimicrobial activities of macrophages and can grow within these cells. Macrophages have other immune surveillance roles including the acquisition of cytosolic components from multiple types of cells. We hypothesized that intracellular pathogens that can replicate within macrophages could also exploit cytosolic transfer to facilitate bacterial spread. We found that viable *Francisella tularensis*, as well as *Salmonella enterica* bacteria transferred from infected cells to uninfected macrophages along with other cytosolic material through a transient, contact dependent mechanism. Bacterial transfer occurred when the host cells exchanged plasma membrane proteins and cytosol via a trogocytosis related process leaving both donor and recipient cells intact and viable. Trogocytosis was strongly associated with infection in mice, suggesting that direct bacterial transfer occurs by this process in vivo.

## Introduction

All intracellular pathogens enter and replicate inside some type of host cell. At the earliest stage of disease only a limited number of host cells will be infected. In order to successfully continue propagation intracellular pathogens must continually infect new susceptible cells. Most of these organisms are thought to infect a cell, replicate, re-enter the extracellular space and start the process over again. However, re-entering the extracellular space exposes the pathogen to antibodies, complement, and other extracellular antimicrobial factors that can inhibit their growth or block their entry into new cells. It is therefore not surprising that certain intracellular pathogens have evolved mechanisms to transfer directly from infected to uninfected cells. The majority of intracellular bacterial pathogens that are known to transfer directly from cell to cell do so through a process known as actin based motility. While there are modest variations in the specific mechanisms employed by individual species, in general the process is pathogen driven through the expression of effector proteins that nucleate and polymerize host cell actin in a manner that physically propels the bacteria into a neighboring cell (*Ireton, 2013*).

There are, however, natural host cell processes that transfer cytosolic material that could be exploited by intracellular pathogens to facilitate direct cell to cell spread. Many recent studies have demonstrated that host cells can exchange cytosolic or membrane materials with neighboring cells through contact-dependent mechanisms (*Joly and Hudrisier, 2003*; *Rogers and Bhattacharya, 2013*). The exchange of cytosolic components occurs in different contexts across a wide range of distinct cells types, and there are several morphologically distinct mechanisms that exchange cytosolic material, including nanotubes, gap junctions, cytonemes and synapses (*Onfelt et al., 2006*; *Rogers and Bhattacharya, 2013*; *Kanaporis et al., 2011*; *Roy et al., 2014*). The different exchange mechanism morphologies are associated with the transfer of specific types of material. For example, gap junctions are selectively permeable to ions and small molecules while nanotubes can transfer

**\*For correspondence:**
shaun_steele@med.unc.edu (SS);
kawula@med.unc.edu (THK)

**Competing interests:** The authors declare that no competing interests exist.

**eLife digest** Many of the bacteria that are able to cause disease in humans and other animals are able to grow inside their host's cells. In doing so, these bacteria can avoid being recognized and killed by the host's immune system. However, the ability of the bacteria to grow within the cell is constrained by the limited space and nutrients that are available inside the infected cell. The current theory is that most of these bacteria eventually kill the cell they have infected and are released into the body so that they can infect other host cells. However, since some host cells can exchange material with their neighbors, it is also possible that the bacteria may be able to travel directly between host cells without leaving the safety of the cell environment.

Macrophages are immune cells that patrol the body to identify and destroy damaged host cells, bacteria and other microbes. Macrophages are also able to interact with neighboring healthy cells through a process called trogocytosis ("trogo" is essentially Greek for nibble). During this process, the membranes of the two participating cells briefly fuse and some of the proteins in the membranes are transferred from one cell to the other. Afterwards, the two cells separate but retain the membrane proteins they acquired from the other cell. The purpose of trogocytosis is poorly understood, but it is thought to help the host to develop immune responses against microbes and tumors.

Steele et al. investigated whether infected mouse and human cells can transfer bacteria to healthy macrophages during trogocytosis. The experiments show that two types of bacteria – called *Francisella tularensis* and *Salmonella enterica* – can transfer from infected cells to macrophages via trogocytosis. Furthermore, the cells of mice infected with *F. tularensis* were more likely to undergo trogocytosis, which suggests that the bacterium may promote and use this process to spread throughout tissues in the body.

Together, Steele et al.'s finding show that some bacteria can hijack a naturally occurring cellular process to move between host cells without re-entering the space that surrounds cells, or damaging either the donor or recipient cell. The next steps following on from this work are to find out how much trogocytosis contributes to the spread and progression of disease. A future goal is to understand the molecular mechanism of trogocytosis so it may be possible to develop drugs that can inhibit the spread of the bacteria in patients.

functional organelles from a donor to a recipient cell (*Onfelt et al., 2006*; *Kanaporis et al., 2011*). Certain viral pathogens are known to transfer directly from cell to cell by exploiting one or more of these natural cellular processes. For example, human immunodeficiency virus (HIV) transfers between cells via tunneling nanotubes (*Sowinski et al., 2008*), whereas Human T-lymphotophic virus (HTLV-1) can spread directly from infected to uninfected T-cells through virological synapses (*Igakura et al., 2003*).

The exchange of plasma membrane proteins between eukaryotic cells occurs through a mechanism termed trogocytosis (trogo = Greek for nibble) (*Joly and Hudrisier, 2003*). For trogocytosis to occur two cells form a transient intimate interaction during which the membranes appear to fuse. The cells eventually separate, with each participant cell having acquired plasma membrane components from the partner cell. The transferred membrane proteins retain their orientation and their function until they are recycled via normal membrane turnover. In certain mouse tissues, over half of the cells have undergone detectable trogocytosis at any given time (*Yamanaka et al., 2009*). In immune cells, trogocytosis leads to a variety of acquired functions that likely impact infection and immunity. For example, trogocytosis improves T cell signaling in response to antigens and dendritic cells can activate T cells after acquiring antigens from neighboring cells (*Osborne and Wetzel, 2012*; *Rosenits et al., 2010*; *Wakim and Bevan, 2011*). Trogocytosis has been implicated as a critical factor in several pathologies including cancer biology, tissue engraftment, and vaccination efficacy (*Li et al., 2012*; *Chow et al., 2013*; *Chung et al., 2014*; *Zhang et al., 2008*). Trogocytosis can occur without the transfer of cytosolic material (*Puaux et al., 2006*), but it is unclear if the presumptive transient membrane fusion that occurs during certain types of cytosolic transfer also results in trogocytosis.

Although mammalian cells exchange intracellular and membrane material, there is a major gap in our knowledge about how these transfer mechanisms impact the infectious process of intracellular pathogens. Foreign material including beads and *Mycobacterium bovis* have been shown to transfer directly between macrophages (*Onfelt et al., 2006*). But it is unclear how prevalent these transfer events are, how they influence pathogenesis, and if these transfer events benefit the pathogen (through cell to cell spread), the host (via immune detection or pathogen destruction upon transfer), or some combination of each. These questions are important because direct cell to cell transfer via cytosolic exchange could be a critical part of the infectious life-cycle for certain intracellular pathogens. For example, an estimated 60% of cells infected by HIV occur through direct viral transfer (*Iwami et al., 2015*).

To investigate how cytosolic transfer affects intracellular pathogens, we used the macrophage-tropic, facultative intracellular bacterium *Francisella tularensis* as a model pathogen. Importantly, *F. tularensis* has not been shown to transfer between cells and lacks homologs of proteins that other bacteria use to transfer between cells. But the rapid spread of *F. tularensis* to new cells and cell types during infection suggests that direct cell to cell transfer may occur (*Hall et al., 2008*; *Roberts et al., 2014*; *Lindemann et al., 2011*). Here, we demonstrate that live *F. tularensis* bacteria transfer directly from infected cells to macrophages via a contact and cytosolic exchange dependent mechanism. Direct bacterial transfer appears to occur frequently both in vitro and in a mouse infection model. Bacterial transfer was cell type specific and correlated strongly with trogocytosis, specifically the exchange of functional major histocompatibility complex I (MHC-I). Lastly, we observed similar transfer events during infections with *Salmonella enterica* or fluorescent beads, suggesting that trogocytosis-associated cell to cell transfer may be a commonly exploited phenomenon.

## Results

### *Francisella tularensis* transfers between macrophages during cytosolic exchange

*Francisella tularensis* is a highly infectious zoonotic bacterial pathogen that is capable of invading and replicating in numerous cell types including, but not limited to, epithelial cells and macrophages (*Hall et al., 2008*). In a mouse model of infection the number of *Francisella* infected cells increases dramatically over a short period of time (*Hall et al., 2008*; *Roberts et al., 2014*). This result suggested to us that *F. tularensis* could spread directly from infected to uninfected cells. To test this hypothesis we monitored GFP- *F. tularensis* infected J774A.1 macrophage-like (J774) cells by live cell imaging. We found that the bacteria transferred from infected to uninfected macrophages upon cell to cell contact (*Video 1* and *2*, *Figure 1A*). After bacterial transfer, both donor and recipient macrophages were typically motile following separation suggesting that both cells remained viable after bacterial transfer (*Videos 1* and *2*, *Figure 1A*). From these results, we concluded that *F. tularensis* bacteria could transfer directly between J774 cells without entering the extracellular space.

We next developed a flow cytometry assay to quantify the transfer event to determine if the direct transfer of bacteria from cell to cell that we observed by live cell imaging occurred at a sufficient frequency to be biologically relevant. In these experiments, we infected cells with *F. tularensis*, added the antibiotic gentamicin to kill extracellular bacteria and allowed the intracellular bacteria to proliferate for 18 hr. We then added uninfected recipient cells labeled with Cell Trace Red to these infected cells. The mixed cell population was co-cultured for 6 hr in the presence of gentamicin and then quantified infected recipient cells by flow cytometry based on double staining for Cell Trace Red and intracellular bacteria. Under these conditions *F. tularensis* transferred from infected to uninfected recipient J774 cells, mouse bone marrow derived macrophages (BMDMs) and primary human monocyte derived macrophages (hMDMs) (*Figure 1B*).

To validate that these transfer events occurred through direct cytosolic exchange rather than from extracellular bacteria, we tracked the transfer of *F. tularensis* with the cytosolic dye calcein-AM which becomes membrane impermeable after entering the cell. *F. tularensis* transfer strongly correlated with the transfer of the cytosolic dye between cells (*Figure 1C,D*). These data indicate that the majority of newly infected cells were infected through the exchange of cytosolic material rather than from extracellular bacteria.

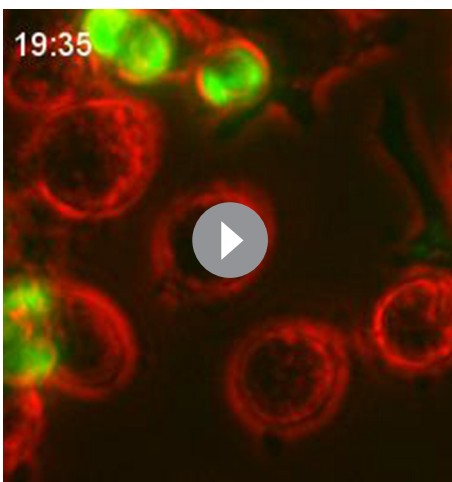

**Video 1.** *F. tularensis* bacteria transfer from infected macrophage to neighboring cells. Time lapse video of an *F. tularensis* infected J774 macrophage (top middle in opening frame) that migrates to neighboring cells, infects those macrophages, and then migrates away. *F. tularensis* is depicted in green, bright field in red. The time is hours: minutes post inoculation. Images were acquired every 5 min.

It is important to note that there are very few extracellular *F. tularensis* bacteria in media containing gentamicin. We found that the number of cells infected at each tested interval was significantly higher than the total number of extracellular bacteria in a milliliter of media (*Figure 1—figure supplement 1A*). Further decreasing the number of extracellular bacteria by inhibiting infected cell lysis through apoptosis or necrosis had no detectable effect on the number of cells infected (*Figure 1—figure supplement 1B*). These results further support our conclusion that most of the newly infected macrophages become infected via cell to cell spread.

## Bacterial transfer requires cell to cell contact

The live cell images suggested that *F. tularensis* was transferred upon cell to cell contact with no obvious infection of new cells through bacteria – containing exosomes. We verified that cell contact dependent transfer was the predominant method for new cells to become infected in the following experiment. We compared the amount of cells that became infected when the cells could come into physical contact with cells that were physically separated by a bacteria permeable membrane. After 12 hr of co-incubation with no antibiotics in the media, there was a roughly 17% increase in infected cells that were able to physically touch compared to a roughly 4% increase in infected cells that were separated by the membrane (*Figure 1E*, Experimental design in *Figure 1—figure supplement 2A*). The reciprocal set-up yielded similar results (data not shown). From these results, we conclude that the majority of bacterial transfer events that we observed occurred from contact-dependent transfer.

## Viable bacteria transfer between cells to propagate infection

*F. tularensis* bacteria transferred between cells, but it is unclear if the transferred bacteria were viable or could sustain growth in the newly infected cells. To assess bacterial viability after transfer, we permeabilized the host cell and measured bacterial viability by propidium iodide exclusion. The percent of viable bacteria was similar between the donor and recipient populations, indicating that bacteria were not killed during transfer between cells (*Figure 2A,B* and *Figure 2—figure supplement 1*).

The best way to accurately assess the contribution of cell to cell transfer on the overall intracellular proliferation of *F. tularensis* would be to compare bacterial growth between conditions that permit and inhibit cell to cell transfer. We therefore screened numerous membrane altering factors for an inhibitor that blocked bacterial transfer and found that the addition of soy

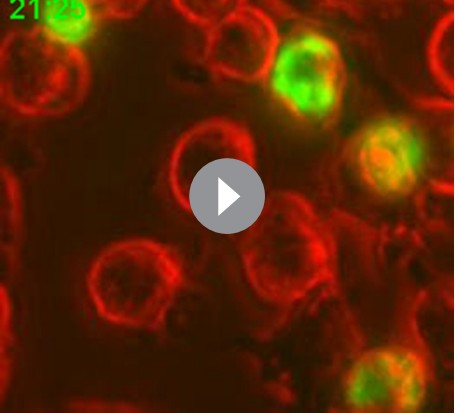

**Video 2.** *F. tularensis* bacteria transfer from infected macrophage to neighboring cells. Time lapse video from an experiment separate from *Video 1* depicting an *F. tularensis* infected J774 macrophage migrating toward neighboring cells, infects those macrophages, and then migrating away. *F. tularensis* is depicted in green, bright field in red. The time is hours: minutes post inoculation. Images were acquired every 5 min.

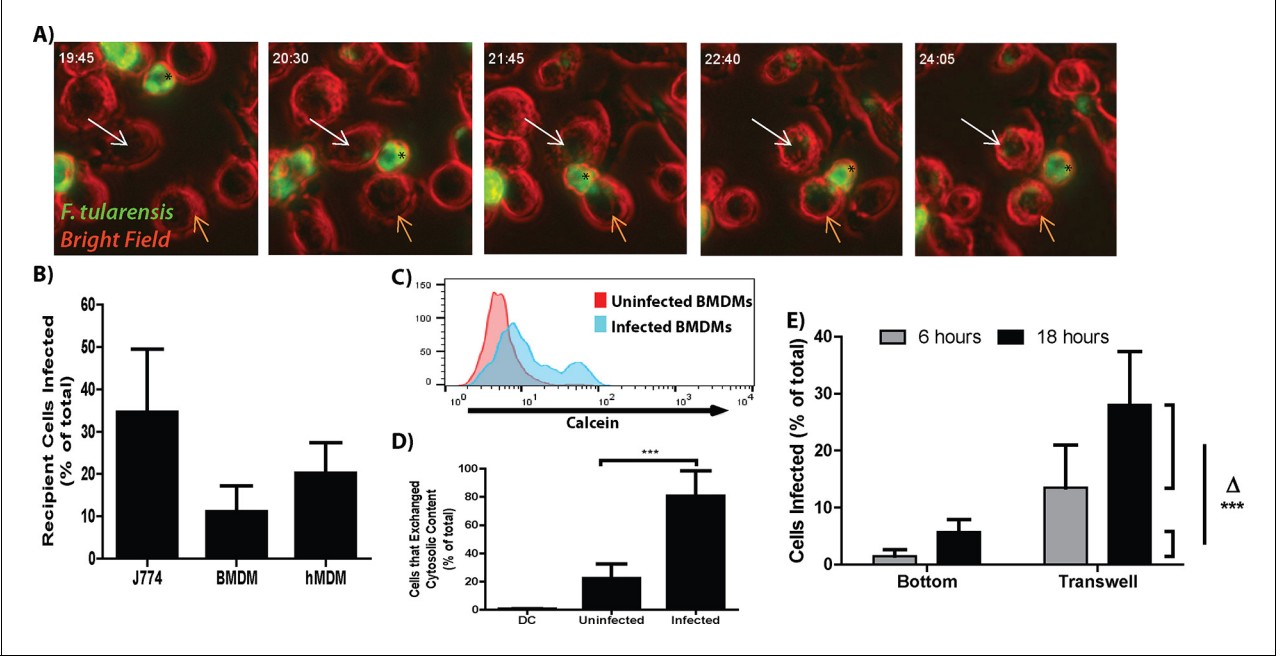

**Figure 1.** *F. tularensis* transfers between macrophages during cytosolic transfer. (**A**) Representative panels from live cell imaging of *F. tularensis* infected J774 cells transferring bacteria. Time- hour: minutes post inoculation; * - initially infected cell; White arrow- first bacterial transfer event; Orange arrow- second bacterial transfer event. Movie available as *Video 1*. (**B**) The proportion of recipient macrophages infected after a 6 hr co-incubation with infected cells of the same type (3 independent experiments performed in triplicate). (**C**) A representative histogram of the amount of calcein that transferred to recipient cells ($\log_{10}$ fluorescence) after 6 hr co-incubation. (**D**) The percent of infected or uninfected cells that exchanged cytosolic content (positive for both Cell Trace Red and calcein) after 6 hr co-incubation. The uninfected population represents cells in the infected well that did not become infected. DC refers to a doublet control (2 independent experiments performed in triplicate) (**E**) Bacterial transfer to uninfected cells is significantly higher with direct cell to cell contact. Infected BMDMs on a transwell filter were suspended over uninfected BMDMs. The percent of total cells infected on the bottom chamber (bottom) and top filter (transwell) were determined by FACS 6 and 24 hr after suspending the transwell over uninfected cells. Side brackets indicate the change in numbers of infected cells in each chamber from 6 to 24 hr. Transfer to BMDMs separated from the initially infected cells was significantly lower than transfer to BMDMS in contact with the infected cell population. (3 independent experiments performed in triplicate). (Mean +/- SD). (***$p<0.001$)

The following figure supplements are available for figure 1:

**Figure supplement 1.** The extracellular space is not a major source of infectious bacteria.

**Figure supplement 2.** Experimental design and bacterial motility for transwell assay.

lecithin after infecting BMDM effectively blocked the transfer of *F. tularensis* to uninfected cells (*Figure 2C*). To test if the transferred bacteria could sustain infection, we infected approximately 1% of a BMDM population then added soy lecithin to inhibit cell to cell contact dependent bacterial transfer. We monitored bacterial viability over 3 days. Cells that cannot directly transfer bacteria will lyse after peak infection (~24 hr), releasing their bacteria into antibiotic containing media. If the bacteria in the untreated cells survive transfer and are able to proliferate, bacterial viability should be higher in these samples than samples treated with soy lecithin at time points after peak infection. We found that both wild-type and transfer-inhibited cells reached peak infection at 24 hr post inoculation, but the lecithin treated samples had significantly fewer viable bacteria compared to untreated samples at 48 and 72 hr post inoculation (*Figure 2D*). Thus, *F. tularensis* exploits cell to cell transfer to extend infection by invading and replicating in previously uninfected cells without entering the extracellular space.

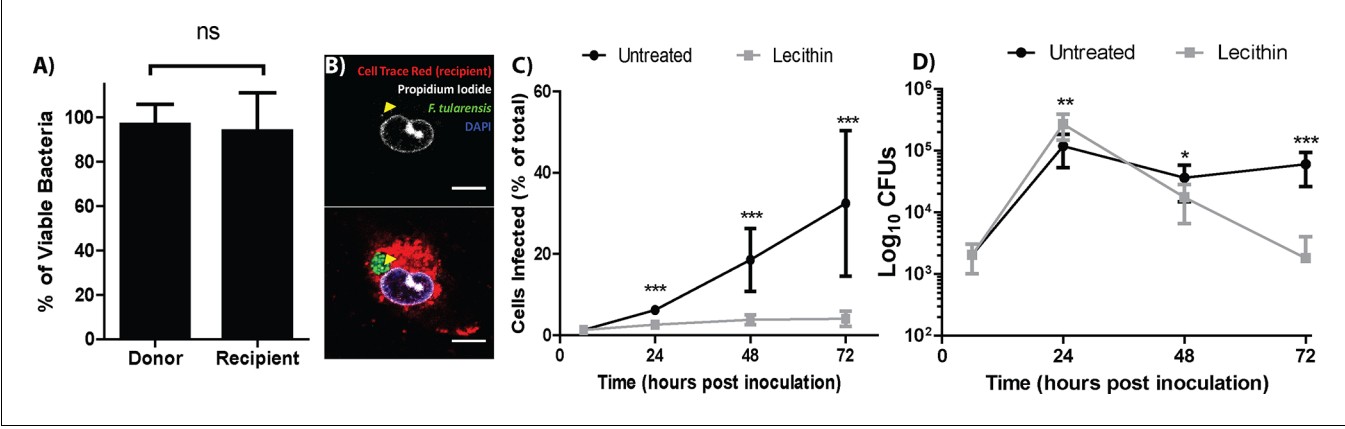

**Figure 2.** Live bacteria transfer to macrophages during bacterial transfer. (**A**) The percent of viable bacteria (propidium iodide negative) in donor and recipient BMDMs (2 independent experiments, 50 fields of view each) (**B**) Micrographs of propidium idodide treated permeabilized *F. tularensis* infected BMDMs. Arrow- propidium iodide positive bacterium. Scale bar- 10 uM. (**C**) The number of cells infected in untreated or soy lecithin treated BMDMs. Infected BMDMs in soy lecithin treated (grey) and untreated (black) were quantified by FACS at indicated times post inoculation. Data are presented as number of infected cells regardless of number of bacteria per infected cell. Soy lecithin was added to the treated populations after initial infection with *F. tularensis.* (**D**) Soy lecithin does not inhibit *F. tularensis* intracellular replication. The number of viable bacteria in untreated or soy lecithin treated BMDMs was quantified at indicated times by dilution plating and calculation of colony forming units. Lecithin was added at 6 hr post inoculation (3 independent experiments performed in triplicate for both lecithin experiments). (Mean +/- SD). (ns p>0.05, *p<0.05, **p<0.01, ***p<0.001).

The following figure supplement is available for figure 2:

**Figure supplement 1.** Propidium iodide can access and bind to dead intracellular bacteria following saponin treatment.

## Bacterial transfer is cell type specific

Many bacterial species transfer from cell to cell through bacteria mediated processes, such as actin based motility. These bacteria can spread between several different host cell types because the transfer mechanisms are driven by bacterial effectors, (*Tilney et al., 1990*; *Makino et al., 1986*; *Heinzen et al., 1993*). To address if *F. tularensis* transfer occurs through a host or bacterial mediated process, we tested *F. tularensis* transfer between different host cell types. Specifically, we compared bacterial transfer between TC-1 epithelial cells, bacterial transfer between macrophages, and bacterial transfer from TC-1 epithelial cells to macrophages. Although *F. tularensis* replicates well in TC-1 epithelial cells (*Fuller et al., 2008*), *F. tularensis* did not detectably transfer from infected to uninfected TC-1 cells (*Figure 3A,B*). However, when we added uninfected BMDMs to the infected TC-1 cells, the BMDMs became infected (*Figure 3C*). The number of infected epithelial cells did not change when BMDMs were added, indicating that the bacteria did not transfer from BMDMs to uninfected epithelial cells (*Figure 3C*). These data indicate that *F. tularensis* transfer is limited to specific recipient cell types and suggest that *F. tularensis* transfer is likely a host mediated process.

## *F. tularensis* does not transfer via previously described bacterial transfer mechanisms

Recipient cell type specificity suggests that *F. tularensis* does not use the transfer mechanisms described in other bacterial pathogens. We further tested this conclusion by comparing *F. tularensis* transfer to other known bacterial mechanisms. We found that *F. tularensis* did not form actin tails that are characteristic of bacterial pathogens such as *L. monocytogenes*, suggesting that *F. tularensis* does not use actin based motility (*Figure 3—figure supplement 1A–C*). Likewise, actin based motility requires continual bacterial protein synthesis (*Tilney et al., 1990*); but, bacterial protein synthesis was not required for *F. tularensis* transfer (*Figure 3—figure supplement 1D*). A proposed alternative form of *F. tularensis* spread is through an autophagy related mechanism termed the *Francisella* containing vacuole (*Checroun et al., 2006*; *Starr et al., 2012*), but inhibiting autophagy with 3-methyladenine (3MA) or using ATG5 knockout BMDMs did not block bacterial transfer (*Figure 3—*

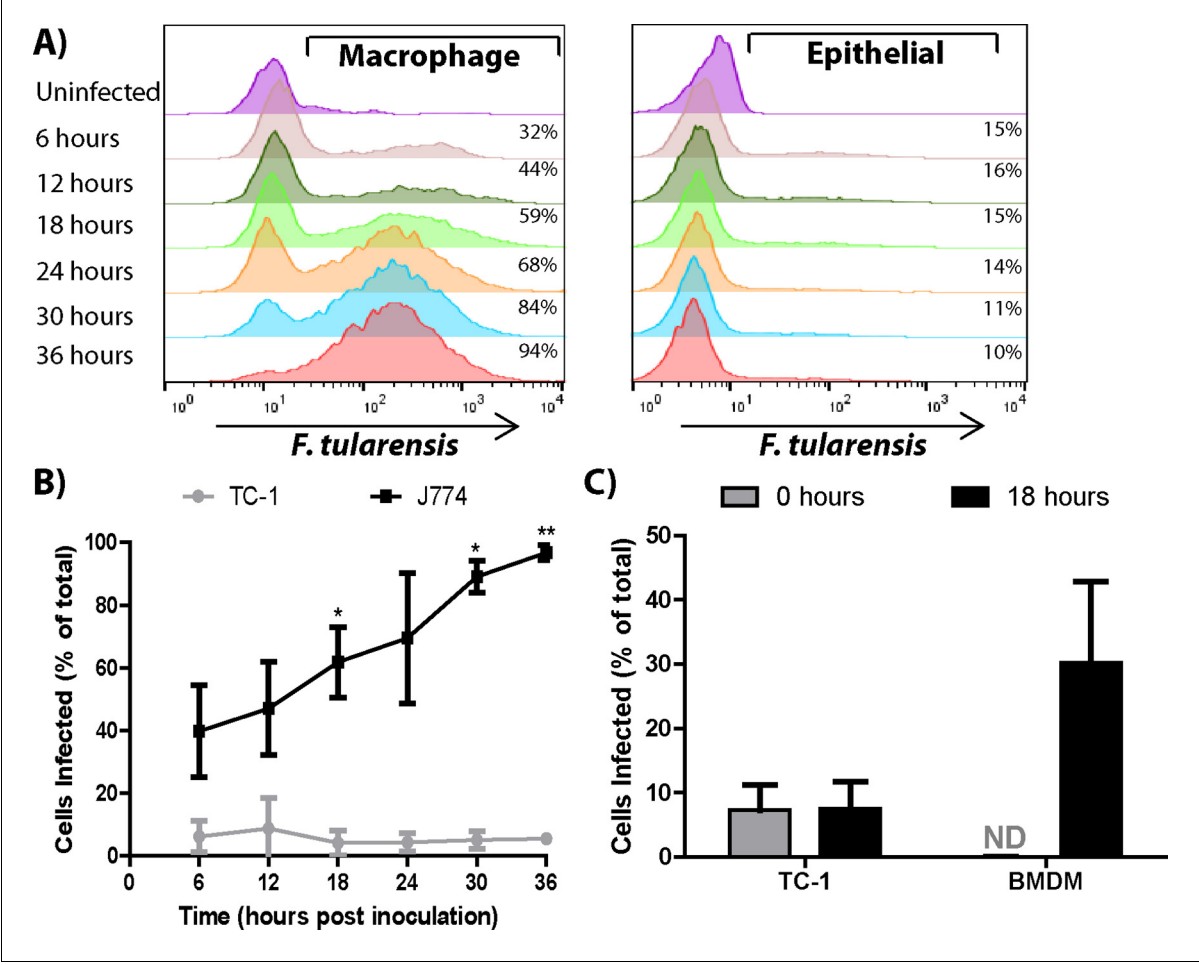

**Figure 3.** Bacterial transfer is cell type specific. (**A**) Percentage of infected J774 macrophages or TC-1 epithelial cells at the indicated time post inoculation (log$_{10}$ fluorescence). (**B**) A compilation of the number of J774 or TC-1 cells infected over time. Statistics represent test for a significant increase in the number of cells infected compared to the previous 6 hr time point. (**C**) TC-1 to TC-1 transfer vs TC-1 to BMDM transfer after a 0 or 18 hr co-incubation. (All results from 3 independent experiments performed in triplicate) (Mean +/- SD). (ns p>0.05, *p<0.05, **p<0.01).

The following figure supplement is available for figure 3:

**Figure supplement 1.** *F. tularensis* does not transfer via actin based motility or autophagy.

*figure supplement 1E*, data not shown). Altogether these data are consistent with *F. tularensis* exploiting host-mediated cytosolic transfer for cell to cell spread.

## Bacterial transfer correlates with trogocytosis

One mechanism for cytosolic exchange observed in cytotoxic T cells (CTL) occurs when pores connecting the cytosol form between the CTL and the target cell (*Stinchcombe et al., 2001*). During this cytosolic intermingling, the cells also exchange specific plasma membrane proteins (*Stinchcombe et al., 2001*). The cell to cell exchange of intact and functional plasma membrane proteins that retain their orientation is termed trogocytosis (*Joly and Hudrisier, 2003*). We noted a similar phenomenon of plasma membrane transfer following bacterial transfer. Newly infected recipient BMDMs frequently acquired plasma membrane proteins as well as cytosolic material from the initially infected cell (*Figure 4A,B*). Interestingly, transferred plasma membrane proteins retained their orientation; so membrane proteins that were surface exposed on the initially infected cell were also surface exposed on the newly infected recipient cell. In the presented images, the cells were stained with a biotin succinimidyl ester prior to mixing the cells with differentially labelled BMDMs. The

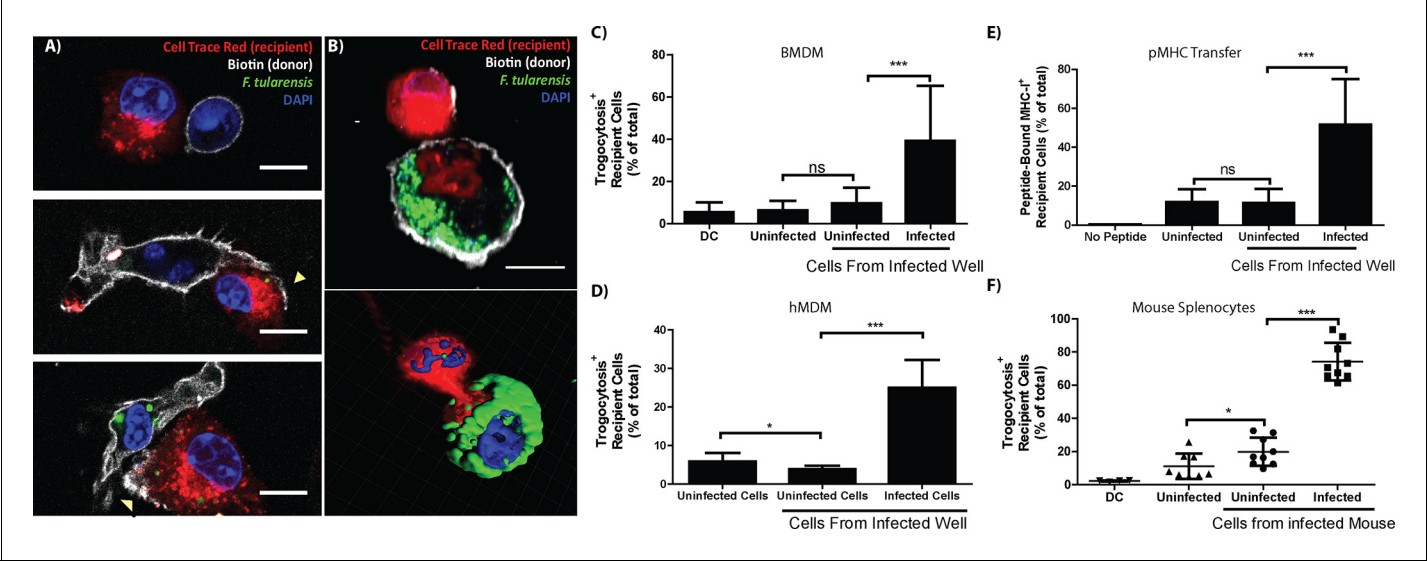

**Figure 4.** Plasma membrane protein transfer correlates with bacterial transfer. (**A**) Fluorescence micrographs of BMDMs before, during, and after trogocytosis. (**B**) A donor [white plasma membrane] and trogocytosis positive recipient BMDM [red cell] exchanging cytosolic material and bacteria. The bottom right panel is a 3D rendering of the Z-stack from the cells in the top panel. Percent of trogocytosis positive recipient cells that are (**C**) BMDMs or (**D**) hMDMs. (**E**) The percent of Balb/c recipient BMDMs that acquired SIINFEKL peptide bound MHC-I from B6 BMDMs. (**F**) The percent of infected splenocytes that underwent trogocytosis in a mouse infection model (8 or 9 mice per group from 4 independent experiments). DC refers to a doublet control. (All tissue culture data are from 3–4 independent experiments performed in triplicate) (Scale bar- 10 um) (Mean +/- SD). (ns p>0.05, *p<0.05, **p<0.01, ***p<0.001)

The following figure supplements are available for figure 4:

**Figure supplement 1.** Plasma membrane protein exchange increases during infection.

**Figure supplement 2.** Trogocytosis does not require *de novo* protein synthesis, but is inhibited by lecithin.

**Figure supplement 3.** Trogocytosis in various cell types in mouse splenocytes.

mixed population was then labeled with a fluorescent conjugated streptavidin immediately before fixation. As a result, the protein must be surface exposed before and after transfer to be labelled (*Figure 4A,B*). These data are consistent with trogocytosis and imply that trogocytosis occurs at the same time as bacterial transfer. Importantly, these results indicate that trogocytosis can be used as a marker for bacterial transfer and differentiate direct *F. tularensis* transfer from more conventional infection mechanisms such as actin based motility or reinfection by extracellular bacteria.

To quantify how often bacterial transfer resulted in detectable levels of trogocytosis, we monitored major histocompatibility complex I (MHC-I) transfer between infected donor and uninfected recipient BMDMs (*Wakim and Bevan, 2011*; *Smyth et al., 2008*). We infected C57BL/6 (B6) BMDMs (MHC-I H2-Kb) and added uninfected Balb/c BMDMs (MHC-I H2-Kd) to the infected B6 cells. After 6 hr of co-incubation, we assayed the Balb/c BMDMs for both *F. tularensis* infection and the acquisition of B6 MHC-I. We found that infection increased the amount of Balb/c BMDMs that acquired B6 MHC-I (*Figure 4—figure supplement 1A*). Likewise, newly infected Balb/c cells were significantly more likely to acquire B6 MHC-I than neighboring Balb/c cells that did not become infected (*Figure 4C*). As with bacterial transfer, trogocytosis did not require *de novo* host or bacterial protein synthesis and was inhibited by treatment with lecithin (*Figure 4—figure supplement 2*). We also observed MHC-I exchange during bacterial transfer when monitoring hMDMs (*Figure 4D*). The surface exposed MHC-I likely remained functional after transfer because it was capable of binding the ovalbumin derived peptide SIINFEKL (*Figure 4E*). Taken together, these data indicate that trogocytosis occurred concurrently with bacterial transfer.

## Infected cells undergo increased levels of trogocytosis in a mouse infection model

We found that trogocytosis is a marker for cell to cell transfer, so we assessed the exchange of plasma membrane proteins in infected splenocytes to track bacterial transfer in vivo. We generated chimeric mice by injecting irradiated F1 B6 and Balb/c mice with wild type Balb/c and transgenic CD45.1[+] B6 bone marrow. In these mice, no cells have genes for both CD45.1 and the MHC-I H2-Kd. Thus, cells must undergo trogocytosis if both CD45.1 and H2-Kd are present on the surface of an individual cell. We infected these mice with *F. tularensis* for 3 days and assayed their splenocytes for infected cells and trogocytosis. Consistent with our in vitro data, *F. tularensis* infection increased trogocytosis (*Figure 4—figure supplement 1B*). Furthermore, infected cells were significantly more likely than uninfected splenocytes from the same mouse to possess both CD45.1 and H2-Kd (*Figure 4F*). Combined with our in vitro data, these results suggest that cell to cell bacterial transfer occurs in a mouse infection model.

The proportion of cells that underwent detectable trogocytosis varied widely between different cell types. Of the cell types we tested, macrophages and monocytes underwent significantly more trogocytosis than dendritic cells or a compilation of all of the other cell types (*Figure 4—figure supplement 3*). These data further indicate that the rate of trogocytosis, and likely bacterial transfer, are cell type specific.

## Trogocytosis-associated bacterial transfer is not restricted to *F. tularensis*

Recipient cell type specificity suggests that trogocytosis-associated bacterial transfer is a host mediated event. If true then other bacterial species that can survive in macrophages should also exhibit cell to cell spread by this mechanism. To test this hypothesis, we assessed bacterial transfer and trogocytosis with *Salmonella enterica serovar Typhimurium (S. typhimurium)* infected cells. Similar to *F. tularensis* cell to cell transfer, *S. typhimurium* infection increased trogocytosis and bacterial transfer correlated with the exchange of MHC-I (*Figure 5A*, *Figure 5—figure supplement 1*).

We also measured the transfer of beads between BMDMs to test if trogocytosis-associated transfer was specific to bacterial infections or occurred with general foreign material. Unlike infections, beads did not increase the level of trogocytosis above baseline (*Figure 5—figure supplement 1*), suggesting that a general bacterial factor increased the rate of trogocytosis and possibly bacterial transfer. But similar to bacterial transfer, the Balb/c macrophages that acquired beads also acquired B6 MHC-I at a significantly higher rate than macrophages that did not acquire beads (*Figure 5B*). The lower rate of total trogocytosis as well as the correlation between trogocytosis and transfer is likely due to phagocytosis of extracellular beads in the well. Taken together, our data demonstrate trogocytosis-associated transfer of intracellular bacteria is a mechanism that potentially any macrophage-tropic intracellular pathogen can exploit.

## Discussion

Our study demonstrates that intracellular bacteria can exploit a host cell cytosolic exchange mechanism to transfer directly from infected cells to macrophages. This cytosolic exchange mechanism is associated with trogocytosis, which is classically defined as the exchange of plasma membrane proteins between two cells. Given that lecithin inhibited both plasma membrane protein and bacterial transfer suggests that trogocytosis and cytosolic exchange are linked with respect to the mechanism of bacterial cell to cell transfer. The bacteria are viable after transfer and can use direct cell to cell transfer to sustain infection without entering the extracellular space. During infections, trogocytosis-associated transfer is a likely mechanism for *F. tularensis* dissemination. Alveolar macrophages are essentially the only cell type initially infected by *F. tularensis* following intranasal inoculation in mice (*Roberts et al., 2014*). Prior to peak infection dendritic cells become infected with *F. tularensis* and these newly infected cells traffic to the draining lymph node (*Bar-Haim et al., 2008*). We postulate that direct cell to cell transfer is the mechanism for the bacteria to transfer from alveolar macrophages to these dendritic cells. Subsequent transfer events may also contribute to systemic dissemination from the lymph node.

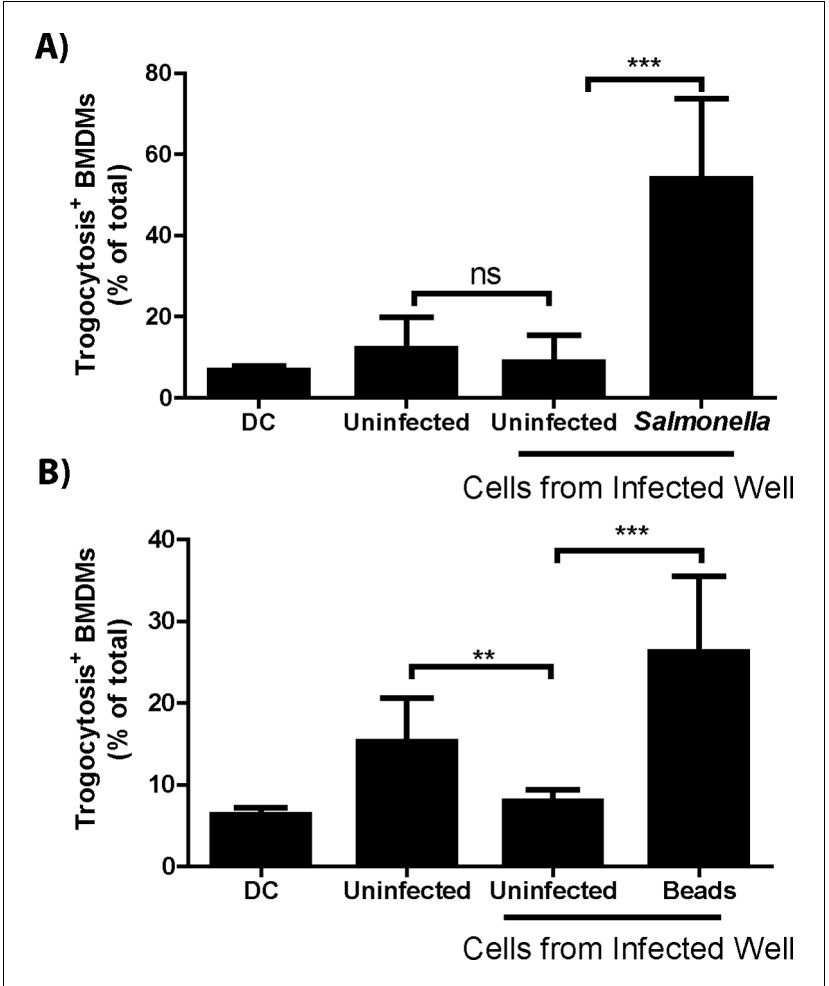

**Figure 5.** Trogocytosis – associated bacterial transfer is not restricted to *F. tularensis*. The percent of recipient cells that underwent plasma membrane protein transfer in response to (**A**) *Salmonella typhimurium* or (**B**) fluorescent beads. The recipient BMDMs that acquired bacteria or beads grouped separately from recipient cells in the same well that did not acquire foreign material. DC refers to a doublet control. (All results from 3–4 independent experiments performed in triplicate) (Mean +/- SD). (ns p>0.05, **p<0.01, ***p<0.001)

The following figure supplement is available for figure 5:

**Figure supplement 1.** Plasma membrane protein exchange increases during infection.

An important question is why certain bacterial species evolved mechanisms to transfer between cells if bacteria can already transfer through cytosolic exchange. In a confluent monolayer of primary cells, only 10–20% of macrophages became infected with *F. tularensis* via trogocytosis-associated transfer in a 6 hr interval (*Figure 1B*). The efficiency of transfer is probably far lower in an infected host because fewer cells are infected and the concentrations of macrophages are much lower. Bacteria that encode mechanisms such as actin based motility likely increase the rate of cell to cell spread. Additionally, certain bacterial species use actin based motility to transfer between epithelial and endothelial cells, whereas *F. tularensis* does not (*Makino et al., 1986*; *Heinzen et al., 1993*; *Reed et al., 2014*). It is possible these transfer mechanisms evolved so that these bacteria can transfer between cells types that do not undergo trogocytosis-associated transfer or to increase the rate of transfer.

Bacterial transfer through a trogocytosis-like process is limited to specific recipient cell types, suggesting that this transfer mechanism is a host mediated event. The spread of bacteria aids in

expanding the replicative niche and possibly dissemination. So why would the host undergo a process that is so potentially deleterious? In cancer biology, trogocytosis of pMHC-I and pMHC-II results in a cytotoxic T cell response to the tumor (*Dolan et al., 2006*; *Zhang et al., 2008*). The immune system may use a similar tactic during infection. Epithelial cells transfer whole antigen to macrophages and dendritic cells to initiate a T cell response via cytosolic exchange (*Ramirez and Sigal, 2002*). Trogocytosis-associated transfer may be the mechanism for this antigenic dissemination. Our work focused on macrophages because *F. tularensis* primarily infects this cell type. However, macrophages are not an ideal cell type to stimulate an adaptive immune response. Future work on trogocytosis-associated transfer in dendritic cells or in the context of adaptive immunity may reveal an important immunological function for this process.

An unexpected observation from our work was that the rate of trogocytosis increased during infection in primary cells and in a mouse infection model (*Figure 4—figure supplement 1*, *Figure 5—figure supplement 1*). These results suggest that infected cells expressed a signal of some kind that initiated, enhanced, or stabilized trogocytosis. This signal is likely not soluble or generalizable because the frequency of trogocytosis only increased in infected cells, not neighboring uninfected cells in the same experimental sample. Trogocytosis is an important immunological process with broad consequences on host engraftment, vaccine efficacy, immune regulation and tumor recognition (*Chow et al., 2013*; *Zhang et al., 2008*; *Li et al., 2012*; *Gu et al., 2012*). Our results indicate that a bacterial stimulus increases the rate of trogocytosis. Future efforts to discern the bacterial products or processes responsible for trogocytosis up-regulation may lead to a specific tool to manipulate trogocytosis.

Trogocytosis-associated bacterial transfer is likely beneficial to both the host and macrophage-tropic pathogens depending on the context. Based on our results, we postulate that trogocytosis-associated transfer benefits certain pathogens early during infection by enhancing dissemination, but could also help initialize or propagate a T cell response that eventually clears the pathogen. Future studies on how this process impacts pathogenesis will likely improve our understanding of how bacteria spread in the host and how the innate immune system acquires antigen to initiate the adaptive immune response.

## Materials and methods

### Bacterial growth

*Francisella tularensis* subsp. *tularensis* Schu S4 was obtained from Biodefense and Emerging Infectious Research Resources Repository (BEI Resources) and *Francisella tularensis* subsp. *holartica* live vaccine strain (LVS) expressing GFP was generated as described (*Hall et al., 2008*). Schu S4 was used for all experiments shown except live cell imaging. Prior to infection, *F. tularensis* was grown overnight in Chamberlin's defined media. *L. monocytogenes* and *S. typhimurium* were grown overnight in Luria broth.

### Antibodies and critical reagents

The clone numbers for the antibodies used in these experiments: anti-*F. tularensis* lipopolysaccharide (1.B.288, US Biologicals; Salem, MA), anti-MHC I H2-Kd (SF1-1.1.1, eBioscience; San Diego, CA), anti-MHC I H2-Kb (AF6-88.5.5.3, eBioscience), anti-MHC I HLA-A2 (BB7.2, eBioscience), anti-CD45.1 (A20, eBioscience), anti-CD45 (30-F11, eBioscience), anti-MHC I H2-Kb-SIINFEKL (25-D1.16, eBioscience).

The catalog number and company for critical reagents used in these experiments: Cell Trace Red DDAO-SE (C34553, Life Technologies; Carlsbad, CA), Calcein-AM (C3099, Life Technologies), Soy Lecithin (Cas number 8002-43-5, Acros; Waltman, MA), phalloidin (A22287, Life Technologies), 3 um pore Transwells (3402 Costar; Corning, NY), gentamicin (15750-060, Gibco; Carlsbad, CA)

The beads (M-1002-010, Solulink; San Diego, CA) used in these experiments were labeled with AF488 succinimidyl ester (A-20100, Life Technologies) to make fluorescent beads.

### Cell culture

TC-1 lung epithelial cells (ATCC CRL-2785; Manassas, VA) were maintained in RPMI supplemented with sodium pyruvate, L-glutamine and non-essential amino acids in 10% fetal bovine serum (FBS,

Gibco). J774A.1 macrophage-like cells (ATCC TIB-67) were maintained in DMEM containing 10% FBS supplemented with sodium pyruvate and L-glutamine. All cell types were kept at 37°C and 5% CO2. All cell types were checked for proper morphology prior to every experiment and consistently monitored for changes in cell replication that might indicate *Mycoplasma* contamination.

For the BMDM, TC-1 and J774 transfer experiments, cells were seeded the night before the experiment at 250,000 cells per well in non-tissue culture treated 12 well dishes or 500,000 cells per well in a 6 well dish on coverslips for microscopy. BMDMs were generated as previously described (*Mortensen et al., 2010*). Unless otherwise indicated, cells were infected with *F. tularensis* at a multiplicity of infection (MOI) of 100 , *S. typhimurium* at an MOI of 10 or beads at an MOI of approximately 1. 10 ug/ml of gentamicin was added at 2 hr post inoculation when BMDMs or J774s were infected or 3 hr post inoculation for TC-1 cells. For co-incubation experiments, the indicated recipient cell type was added to the infected cells at 18 hr post inoculation and harvested at 24 hr post inoculation unless otherwise indicated.

Primary human monocyte derived macrophages were generated by acquiring human blood in heparin tubes and isolating the peripheral blood mononuclear cells (PBMC) and serum on a ficoll gradient. The cells were plated in Iscove's modified Dulbecco's medium (IMDM) for 2 hr. The non-adherent cells were washed away and the media was replaced with IMDM containing 5% autologous human serum. Primary human cells were cultured for 7 days prior to infection. The blood was isolated from several healthy volunteers who gave informed, written consent following an approved protocol by the Institutional Review Board for human volunteers at the University of North Carolina at Chapel Hill. Blood was obtained specifically for these experiments. Different donors were used for each experiment.

The infected cells were seeded onto a coverslip for all experiments involving primary human cells. The coverslip was inverted in a well of uninfected cells so that the infected cells were in contact with the uninfected cells. The reciprocal setup was used for TC-1 to BMDM transfer experiments. Other methods to transfer the cells resulted in large amounts of cell lysis.

## Live cell imaging

For live cell imaging, J774 cells were infected at an MOI of 500 with GFP-expressing *F. tularensis* LVS bacteria in a synchronous infection. Briefly, the J774 cells were chilled on ice for 30 min, the media was exchanged with media containing the bacteria, centrifuged for 5 min and then the bottom of the plate was placed in a 37°C water bath for 2 min. The cells were incubated for 15 min in an incubator at 37°C and 5% carbon dioxide and then the media was replaced with media containing gentamicin. The cells were then imaged every 5 min for 24 hr using a 40x objective on an Olympus IX70 microscope in a temperature and carbon dioxide contained chamber. All data were analyzed using ImageJ (*Schneider et al., 2012*)

## Bacterial transfer inhibition assay

BMDMs were seeded at 500,000 cells the night before infection. Cells were infected with an MOI of 0.5 bacteria and 10 ug/ml of gentamicin was added at 2 hr post inoculation. 0.5 mg/ml of soy lecithin (Acros) was added with gentamicin at 6 hr post inoculation. 50% of each sample was used for viable bacteria quantification through serial dilutions and plating on chocolate agar. The remaining 50% of the sample was used to determine the number of cells infected as previously described.

Soy lecithin is a common emulsifier often used in food preparation that significantly blocks bacterial transfer (*Figure 2C*) and trogocytosis (*Figure 4—figure supplement 2*). We were unable to ascertain the precise mechanism behind this inhibition, but suspect that it is due to its properties as an emulsifier because other complex phospholipid mixtures such as bovine lung surfactants (Survanta) also decreased bacterial transfer, albeit to a lesser extent (data not shown). Treating infected cells with individual phospholipid components of soy lecithin, such as phosphatidylcholine, did not affect bacterial transfer (data not shown).

## Flow cytometry assays

When analyzing surface markers (CD45, H2-KD, H2-KB, or H2-KB-SIINFEKL), cells were stained in the wells in which they were infected. We added 2.4G2 cell supernatant (Fc blocking buffer) to infected cells for 5 min. The 2.4G2 supernatant was removed and antibodies were added. After 5 min, the

cells were washed twice in PBS containing 2% fetal bovine serum (FBS), re-suspended, and fixed in 4% paraformaldehyde.

*F. tularensis* within infected cells were detected by permeabilizing the plasma membrane with 0.1% saponin (Millipore) in PBS and 2% FBS (Gibco). The cells were stained with an anti-*F. tularensis* lipopolysaccharide antibody (US biological) conjugated to either Pacific blue, AF488, or AF647 by combining the antibody with a succinimidyl ester of the dye. The conjugated antibody was separated from unbound dye by a 30,000 molcular weight filter and repeated washes with PBS and glycine. Conjugation efficiency was then assayed for each batch. We were able to detect bacteria at 1 hr post-inoculation when as few bacteria as 1 bacteria per cell were present (data not shown).

We stained for both extracellular and intracellular bacteria and found that 1% or less of the infected BMDMs were positive due to surface bound extracellular bacteria (data not shown). Due to the low number of false-positive events, we did not stain specifically for extracellular bacteria in the majority of assays so that we could minimize spectral overlap of our panel.

All mouse plasma membrane protein transfer experiments included a doublet control. Uninfected cells from both populations were each stained with all antibodies. Each population was removed from the plate and combined in 4% paraformaldehyde.

## Transfer of cytosolic dyes

BMDMs were infected for 18 hr and then stained with calcein-AM following the manufacturer's protocol (Invitrogen, Grand Island, NY). Uninfected BMDMs were concurrently stained with Cell Trace Red (Invitrogen) following the manufacturer's protocol. The different populations were either fixed immediately for controls or combined and co-incubated for 6 hr. The cells were then stained for *F. tularensis* as described above.

## Transwell assay

The day before infection, BMDMs were seeded either in a 12 well plate or in the chamber of 12 mm, 3.0 uM pore transwell. Each chamber (transwell and plate) was kept separate. One chamber per pair was infected and 10 ug/ml of gentamicin was added at 2 hr post inoculation to kill any extracellular bacteria. At 6 hr post-inoculation, the gentamicin was removed and the infected and uninfected chambers were combined. We then separated and harvested each chamber at either 6 or 18 hr post inoculation.

To test for bacteria traversing the membrane, we combined the chambers, added bacteria directly to the media of the indicated chamber (MOI 100) and tested for the number of infected cells in each chamber 2 hr later (*Figure 1—figure supplement 1B*).

## Extracellular bacterial enumeration

BMDMs were infected for 2 hr and then gentamicin was added. At 6 hr post inoculation, the media was exchanged for media with or without gentamicin. At 6 hr intervals, the cells were harvested and stained for intracellular *F. tularensis* and the media was serially diluted and plated on chocolate agar. To approximate the number of cells infected every 6 hr, we used the change in infection percentage between intervals and assumed the number of BMDMs doubled overnight. We made this assumption based on previous observation of chromosomal segregation in infected BMDMs by microscopy (data not shown). The conclusions, however, would remain the same even if no cell division is assumed.

## Cell death and autophagy inhibition

BMDMs were infected and gentamicin was added at 2 hr post inoculation. At 6 hr post inoculation, the media was exchanged for media containing gentamicin and the indicated treatment. Z-Vad (OMe)-FMK (Cayman Chemical, Ann Arbor, MI ) was used at 20 uM and Necrostain-1 (Cayman Chemical) at 10 uM. At 6 or 24 hr, samples were harvested and analyzed for intracellular bacteria.

Autophagy inhibition experiments were performed in the same manner, with 10 µM 3-methyladenine (Cayman Chemical) added at 18 hr post inoculation.

## Quantification of live and dead intracellular bacteria

Cell Trace Red BMDMs were added to GFP-expressing *F. tularensis* infected BMDMs 18 hr post inoculation. At 24 hr post inoculation, the cells were treated with 0.1% saponin in PBS and 2% FBS for 15 min at room temperature (wash buffer). 3 uM propidium iodide (PI) was added to the cell for 12 min in wash buffer. The cells were washed 3 times and then fixed in paraformaldehyde. GFP positive, PI negative live bacteria and GFP, PI double positive dead bacteria were enumerated using fluorescence microscopy. To ensure that PI could access and bind to dead bacteria following saponin treatment BMDMs infected with the GFP expressing intracellular growth impaired mutant △FTT_0924 (*Brunton et al., 2015*) were subjected to the same procedure (*Figure 2—figure supplement 1*).

## Epithelial to BMDM transfer

TC-1 epithelial cells were infected for 6 hr as described above. A cover slip seeded with BMDMs was inverted on top of the infected TC-1 cells and the cells were co-incubated for 18 hr in media containing gentamicin. At 24 hr post inoculation, the slide was removed and the TC-1 and BMDM cells that migrated from the cover slip to the bottom of the plate were stained for CD45 to determine cell type, fixed, and then stained with *F. tularensis* LPS antibody as described above. The 0 hr co-incubation represents TC-1 cells that were infected for 24 hr but did not have BMDMs added to the well.

## Mice

All mice were obtained from Jackson Laboratory (Bar Harbor, ME) and were housed in specific pathogen free housing at the University of North Carolina- Chapel Hill. All mouse experiments were performed under approved protocols from the University of North Carolina- Chapel Hill Institutional Animal Care and Use Committee. All mice used were female. The age of mice for bone marrow macrophage production varied (6 weeks to 6 months old). All mice used to generate chimeric mice were 6 weeks old at the time of irradiation or bone marrow harvest.

## Bone marrow chimera mouse experiment

F1 mice from a mating of C57Bl/6 and Balb/c mice were irradiated with 1000 cGY using an X-ray irradiator. About 5 hr after irradiation, the irradiated mice were reconstituted by intravenous injection of 10 million T cell depleted bone marrow cells per mouse (T cells depleted using Miltenyi CD3e Microbead Kit following the manufacturers protocol). The bone marrow cells were approximately a 1:1 mixture of cells from wild-type Balb/c mice and CD45.1 C57bl/6 mice (B6.SJL-PTprc[a] Pepc[b]/ BojJ). No blinding was performed in these studies.

Five to seven weeks after irradiation, half of the bone marrow chimera mice in each irradiation group were infected intranasally with approximately 500 colony forming units of GFP-expressing *F. tularensis* Schu S4. Mice were randomly assigned to each group. At 3 days post inoculation, the spleens were harvested and made into a single cell suspension. The cells were treated with ammonium chloride lysing buffer to removed red blood cells. The splenocytes were then stained with anti-CD45.1 and anti-H2-KD (Balb/c MHC I) antibodies, washed, fixed in 4% paraformaldehyde, stained for intracellular *F. tularensis* and analyzed by flow cytometry.

## Plasma membrane protein transfer (Trogocytosis) assays

C57BL/6 BMDMs were infected and gentamicin was added at 2 hr post inoculation. At 18 hr post inoculation, Balb/c BMDMs were added to the infected B6 cells in the presence of gentamicin. For select experiments, 0.5 ug of the ovalbumin peptide SIINFEKL (ova 257–264) (AnaSpec Inc) was also added at 18 hr post inoculation. At 24 hr, the cells were stained and harvested for flow cytometry. All flow cytometry experiments included a doublet control, where stained and paraformaldehyde fixed B6 and Balb/c cells were mixed at approximately a 1 to 1 ratio with a similar cell concentration as the rest of the samples. The doublet control sample represents the background level of false positives for plasma membrane protein transfer due to doublets.

Experiments with *S. typhimurium* or magnetic beads were performed by infecting B6 BMDMs with an MOI of 10 GFP expressing *S. typhimurium* bacteria or an MOI of 1 streptavidin coated magnetic bead (Solulink) conjugated to AF488. At 2 hr post inoculation, the cells were washed and

media containing 25 ug/ml of gentamicin was added. At 10 hr, Balb/c BMDMs were added and the samples were harvested at 16 hr. The samples were surface stained as previously described.

For microscopy, infected BMDMs were biotinylated at 18 hr post inoculation (Thermo Scientific; EZ-Link Sulfo-NHS-LC-biotin following the manufacturer's protocol). Cell Trace Red labeled BMDMs were added to the infected cells immediately following biotinylation. 1 to 2 hr later, the samples were stained with AF568 or PE conjugated streptavidin, fixed in 4% paraformaldehyde, and mounted using DAPI containing mounting media. Images were acquired using the 63x objective on a Zeiss CLSM 700 Confocal Laser Scanning Microscope. Images were acquired using Zen software (Zeiss). All data were analyzed using ImageJ (*Schneider et al., 2012*). 3D images were generated using Imaris software (Bitplane).

For human samples, HLA-A2 negative, biotinylated MDMs were added to infected HLA-A2+ MDMs at 18 hr post inoculation. The cells were co-incubated for 6 hr and then the recipient cell population was stained with PE-streptavidin and HLA-A2 to assess plasma membrane protein transfer.

## Protein synthesis inhibition

Recipient BMDMs and the indicated treatment (0.1 ng/ml cycloheximide or 50 ug/ml chloramphenicol) were added to infected BMDMs at 18 hr post inoculation. The samples were assessed as described above. At these concentrations, cycloheximide increased the basal rate of plasma membrane protein transfer while chloramphenicol decreased the basal rate of plasma membrane protein transfer.

## Actin localization

Cells were infected with an MOI of 1 for *L. monocytogenes* or 100 for *F. tularensis*. Cells were harvested at 16 hr post inoculation, fixed, permeabilized and stained with AF647 conjugated phalloidin.

## Data analysis

All statistics were performed by a 2 tailed, unpaired Student t-tests using raw data values. Confocal microscopy experiments represent all cells from 100 total fields of view from 2 independent experiments. For statistics, each field of view was treated as an independent sample. Chimeric mouse experiments were performed with 2 mice per group in 4 independent experiments. We estimated the size for these animal studies based on our results in tissue culture. All other experiments were performed in triplicate for each group in at least 3 independent experiments unless otherwise indicated.

## Acknowledgements

We thank B Miller, M Braunstein, N Moorman, and A Richardson for reading the manuscript, R Bagnell and V Madden at the Microscopy Service Laboratory for aid with the microscopy experiments, N Maponga at the Retrovirology Core for drawing blood, C Santos at Animal Studies Core for generating the chimeric mice and the Flow Cytometry Core. We also thank E Miao for the gift of the GFP-expressing *Salmonella typhimurium* and L Lenz for GFP-expressing *Listeria monocytogenes*. This work was supported by National Institutes of Health award AI082870 (TK) and by a University of North Carolina Graduate School Dissertation Award (SS).

## Additional information

### Funding

| Funder | Grant reference number | Author |
| --- | --- | --- |
| National Instutes of Health Award | AI082870 | Thomas H Kawula |
| UNC Graduate School Dissertation Award | | Shaun Steele |

The funders had no role in study design, data collection and interpretation, or the decision to submit the work for publication.

### Author contributions
SS, Conception and design, Acquisition of data, Analysis and interpretation of data, Drafting or revising the article; LR, Acquisition of data, Analysis and interpretation of data, Drafting or revising the article; STB, JB, Acquisition of data, Analysis and interpretation of data; THK, Conception and design, Analysis and interpretation of data, Drafting or revising the article

### Author ORCIDs
Thomas H Kawula, http://orcid.org/0000-0001-7526-5159

### Ethics
Animal experimentation: This study was performed in strict accordance with the recommendations in the Guide for the Care and Use of Laboratory Animals of the National Institutes of Health. All of the animals were handled according to approved institutional animal care and use committee (IACUC) protocols (#13-213.0) of the University of North Carolina.

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
