## [Decision Letter]

Thank you for submitting your work entitled "Trogocytosis-Associated Cell to Cell Spread of Intracellular Bacterial Pathogens" for consideration by *eLife*. Your article has been reviewed by three peer reviewers, and the evaluation has been overseen by a guest Reviewing Editor and Richard Losick as the Senior Editor. The reviewers have discussed the reviews with one another and the guest Reviewing editor has drafted this decision to help you prepare a revised submission.

Summary:

The authors have tested the hypothesis that intracellular pathogens that can replicate within macrophages exploit cytosolic transfer to facilitate bacterial spread. Here they show that viable *Francisella* transfer from infected macrophages to uninfected macrophages along with other cytosolic material through a transient, contact dependent mechanism. The authors describe this as a trogocytosis related process that leaves both donor and recipient cells intact and viable. The mouse results suggest that this process is strongly associated with infection in a mammalian host.

Essential revisions:

1) Address the reviewers’ concerns about the lack of specificity of lecithin. Does lecithin restrict bacteria to the originally infected population? Does lecithin impair plasma membrane protein transfer under the authors' experimental conditions? Alternatively, the use of more specific means to impair trogocytosis would substantiate this aspect of the study.

2) Provide evidence that the propidium iodide assay reliably detects non-viable bacteria and is not subject to a high level of false-negatives due to PI accessibility issues.

3) Rewrite the manuscript to increase clarity. For example, the explanation of experiments is too short and it is often not clear why and how the experiments are being done.

4) Improve the statistical analyses and the descriptions of these analyses.

5) Extend the Introduction.

6) The authors should discuss clearly that the mechanism of bacterial transfer is different from classical trogocytosis in which no cytosolic transfer is described.

[Editors' note: further revisions were requested prior to acceptance, as described below.]

Thank you for resubmitting your work entitled "Trogocytosis-Associated Cell to Cell Spread of Intracellular Bacterial Pathogens" for further consideration at *eLife*. Your revised article has been favorably evaluated by Richard Losick (Senior editor), a guest Reviewing editor, and one of the original reviewers. The manuscript has been improved but there are some remaining issues that need to be addressed before acceptance, as outlined below:

1) It is very important that you show that pretreatment of BMMs with lecithin does not affect uptake, hence replication of *Francisella*. This would greatly strengthen the lecithin data and your conclusion. What one can infer from Figure 2 is that lecithin does not affect *Francisella* replication in the initial population between 6 and 24 h, but it does not verify effects on the original uptake. In addition, for Figure 2, you need to clarify what you are scoring as "infected cells" (these details were not provided): cells with replicating bacteria? Or any cell with at least one bacterium? If infected cells were scored regardless of intracellular bacterial numbers, then it shows a lack of cell-to-cell spread in the presence of lecithin.

2) Please add more details to Figure 2 legend.

3) Please change the presentation of Figure 1 and Figure 3. Since those plots show only two time points, it is incorrect to represent them as "curves". Use a histogram-type presentation with two time points instead.

---

## [Author Response]

*1) Address the reviewers’ concerns about the lack of specificity of lecithin. Does lecithin restrict bacteria to the originally infected population? Does lecithin impair plasma membrane protein transfer under the authors' experimental conditions? Alternatively, the use of more specific means to impair trogocytosis would substantiate this aspect of the study.*

Yes, lecithin restricts bacteria to the originally infected host cell population. This is shown in Figure 2. Lecithin also inhibits plasma membrane protein transfer (trogocytosis). We have added these data to the revised manuscript (Figure 4—figure supplement 2). We screened over 50 compounds looking for non-lethal treatments that inhibited cell-to-cell spread and/or trogocytosis. Soy lecithin was the only non-toxic product we found that inhibited these events without also destroying the host cell cytoskeleton.

*2) Provide evidence that the propidium iodide assay reliably detects non-viable bacteria and is not subject to a high level of false-negatives due to PI accessibility issues.*

We used an intracellular growth impaired mutant △FTT-0924 (Brunton et al. 2015. *Infec* Immun. 83(8):3015-2) as a control for propidium iodide access and binding to dead intracellular bacteria. We added the control data to the revised manuscript (Figure 2—figure supplement 1).

*3) Rewrite the manuscript to increase clarity. For example, the explanation of experiments is too short and it is often not clear why and how the experiments are being done.*

We have extensively rewritten the manuscript with considerable additional experimental detail and explicitly setting the background and rationale for each experiment and result topic section.

*4) Improve the statistical analyses and the descriptions of these analyses.*

The data are now presented as means and standard deviations of the means with corresponding P values.

5) Extend the Introduction.

We have completely rewritten and expanded the Introduction, specifically delineating the known mechanisms and processes of membrane and cytosolic exchange and also including known examples of direct cell to cell transfer by viral pathogens.

*6) The authors should discuss clearly that the mechanism of bacterial transfer is different from classical trogocytosis in which no cytosolic transfer is described.*

We have made every effort to discuss the trogocytosis-associated bacterial transfer mechanism in the context of current trogocytosis definitions. Specifically to this point we added a clear classical definition/description of trogocytosis in the Introduction (third paragraph), and added a comment in the Discussion to the effect that our data suggest that cytosolic exchange may be linked mechanistically to trogocytosis (first paragraph). The latter point has been speculated, but never tested.

[Editors' note: further revisions were requested prior to acceptance, as described below.]

*The manuscript has been improved but there are some remaining issues that need to be addressed before acceptance, as outlined below: 1) It is very important that you show that pretreatment of BMMs with lecithin does not affect uptake, hence replication of Francisella. This would greatly strengthen the lecithin data and your conclusion. What one can infer from Figure 2 is that lecithin does not affect Francisella replication in the initial population between 6 and 24 h, but it does not verify effects on the original uptake.*

Lecithin was added post bacterial uptake, and therefore had no impact on uptake or on the number of initially infected cells. This information is in the “Bacterial transfer inhibition assay” Methods sections (first paragraph), and we have clarified this detail in the Results section (subsection “Viable bacteria transfer between cells to propagate infection”).

*In addition, for Figure 2, you need to clarify what you are scoring as "infected cells" (these details were not provided): cells with replicating bacteria? Or any cell with at least one bacterium? If infected cells were scored regardless of intracellular bacterial numbers, then it shows a lack of cell-to-cell spread in the presence of lecithin.*

Infected cells were identified by flow cytometry as described in the corresponding Materials and methods section. In theory a cell scored as infected could have as few as one bacterium, and we have established that this procedure is sensitive enough to detect such cells (Hall et al. Infection and Immunity 76 (12): 5843-5852. doi:10.1128/IAI.01176-08). We did however set a conservative gate based on mean fluorescence intensity that was significantly above background to ensure that we counted only infected cell. Thus, as suggested by the reviewer, the results confirm our conclusion that lecithin significantly blocked cell-to-cell spread. We have added the infected cell scoring criteria to the figure legend.

2) Please add more details to Figure 2 legend.

We have added more assay and data collection details to the Figure 2 legend.

*3) Please change the presentation of Figure 1 and Figure 3. Since those plots show only two time points, it is incorrect to represent them as "curves". Use a histogram-type presentation with two time points instead.*

We changed the data presentation of Figure 1 and Figure 3 as suggested.